# Protective Immune Responses Induced by an mRNA-LNP Vaccine Encoding prM-E Proteins against Japanese Encephalitis Virus Infection

**DOI:** 10.3390/v14061121

**Published:** 2022-05-24

**Authors:** Tao Chen, Shuo Zhu, Ning Wei, Zikai Zhao, Junjun Niu, Youhui Si, Shengbo Cao, Jing Ye

**Affiliations:** 1State Key Laboratory of Agricultural Microbiology, Huazhong Agricultural University, Wuhan 430070, China; ct@webmail.hzau.eud.cn (T.C.); zhushuo@webmail.hzau.edu.cn (S.Z.); weining@webmail.hzau.edu.cn (N.W.); zikaizhao@hotmail.com (Z.Z.); njj@webmail.hzau.edu.cn (J.N.); youhui@mail.hzau.edu.cn (Y.S.); 2Laboratory of Animal Virology, College of Veterinary Medicine, Huazhong Agricultural University, Wuhan 430070, China; 3The Cooperative Innovation Center for Sustainable Pig Production, Huazhong Agricultural University, Wuhan 430070, China

**Keywords:** Japanese encephalitis virus, mRNA vaccine, prM-E protein, immunogenicity

## Abstract

Japanese encephalitis virus (JEV) is an important zoonotic pathogen, which causes central nervous system symptoms in humans and reproductive disorders in swine. It has led to severe impacts on human health and the swine industry; however, there is no medicine available for treating yet. Therefore, vaccination is the best preventive measure for this disease. In the study, a modified mRNA vaccine expressing the prM and E proteins of the JEV P3 strain was manufactured, and a mouse model was used to assess its efficacy. The mRNA encoding prM and E proteins showed a high level of protein expression in vitro and were encapsulated into a lipid nanoparticle (LNP). Effective neutralizing antibodies and CD8+ T-lymphocytes-mediated immune responses were observed in vaccinated mice. Furthermore, the modified mRNA can protect mice from a lethal challenge with JEV and reduce neuroinflammation caused by JEV. This study provides a new option for the JE vaccine and lays a foundation for the subsequent development of a more efficient and safer JEV mRNA vaccine.

## 1. Introduction

Japanese encephalitis (JE) is a zoonotic disease caused by Japanese encephalitis virus (JEV) which is primarily spread by *Culex tritaeniorhynchus*, and swine is an amplifying host. The genome of JEV is a single-stranded positive-stranded RNA with a total length of about 11 kb and a long open reading frame (ORF) coding a long polyprotein precursor. The virus can not only cause reproductive failure in sows and bring huge losses to the swine industry, but also induce central nervous system symptoms in humans with a fatality rate of about 30%. Close to half of patients who survived have permanent sequelae of the nervous system [1]. According to the World Health Organization, global cases of JE are estimated at 68,000 people each year, while the reality is probably 100,000 cases in Asia [2]. However, there is still no commercial drug that could completely block the infection or replication of JEV in the host or cure JE. Therefore, vaccination is the best preventive measure for swine and humans.

Compared with inactivated vaccines, live-attenuated vaccines based on SA14-14-2 have better immunogenicity and a more convenient immunization program. However, the incidence in adults has increased in recent years, suggesting that a live-attenuated vaccine may not provide long-term effective protection [3,4,5]. In addition, the populations in some Asian countries are worried about the safety of live vaccines and choose inactivated vaccines which are relatively less effective [6]

Other types of JEV vaccines, such as recombinant protein subunit vaccines and DNA vaccines, have been previously reported to show effective protection in mice [7,8]. For example, virus-like particles containing E protein were secreted into the culture medium and immunized mice after purification. The results showed that the vaccine can induce high titers of neutralizing antibodies and completely protect the mice against the JEV challenge [9]. The inoculation of plasmid encoding prM-E protein mixed with colloidal gold could induce a specific anti-JEV response in mice [10]. However, the production processes of recombinant protein subunit vaccines are more complex and easily contaminated by no-target proteins. A traditional DNA vaccine encoding the Japanese encephalitis virus protein showed effective protection in mice [11]. In addition, DNA vaccines based on infectious cloning technology have also been described [12,13]. Nevertheless, DNA vaccines coding associated antigen proteins need to enter the nucleus of the host cell, which may pose safety concerns. Messenger RNA (mRNA)-based vaccines have been rapidly developed in recent years with notable advantages such as no integration into the genome, no anti-carrier immunogenicity and being easier to produce with high purity [14]. As a result of the extremely rapid improvement in delivery systems and modification technology, mRNA vaccines are playing an increasingly critical role in the field of different diseases. In 2017, an mRNA encoding prM and E proteins against Zika virus significantly reduced the antibody-dependent enhancement (ADE) effect in Zika infection and induced high neutralizing antibody titer in mice [15]. In addition, an mRNA vaccine against antigens of tumor cells has been also widely studied and developed. An mRNA vaccine encoding a neoantigen expressed by autologous tumors was reported to treat patients with metastatic gastrointestinal cancer and proved to be safe with a function to stimulate mutation-specific T cell responses [16]. Furthermore, autoimmune diseases are targeted by an mRNA vaccine treatment as well. In 2021, a nucleotide-modified mRNA vaccine encoding self-antigen of diseases was successfully delivered to APC cells in the nonattendance of costimulatory signals, and effectively inhibited diseases, in mice, of multiple sclerosis [17].

In this study, we developed an mRNA vaccine encoding prM and E proteins of JEV and assessed its immunogenicity and protective effectiveness. We found that the mRNA vaccination could induce neutralizing antibodies against JEV and protect mice from the JEV challenge. This research may provide new clues for the development of a more efficient and safer JEV mRNA vaccine.

## 2. Materials and Methods

### 2.1. Viruses, Cells and Plasmids

The JEV P3 strain was stored in our laboratory. HEK-293T (human embryo kidney cells) and BHK-21 (baby hamster Syrian kidney cells) cells were maintained in Dulbecco’s modified Eagle’s medium (DMEM; Gibco, Grand Island, NY, USA) supplement with 10% fetal bovine serum (FBS; Gibco, Grand Island, NY, USA) and incubated at 37 °C in 5% CO_2_. The JEV prM-E genes with a signal sequence peptide were amplified with the primers listed in Table 1 from cDNA and inserted into the pGEM-3zf (+) vector (Promega, Madison, WI, USA), which incorporated the 5′ and 3′ untranslated regions (UTRs) and a poly-A tail.

### 2.2. Generation of mRNA and mRNA-LNP

The mRNA contained 5′ and 3′ UTR and a poly-A tail was produced from a linearized DNA template with a T7 in vitro transcription kit (Cellscript Madison, WI, USA), and pseudouridine was used in place of uridine. Then, the mRNA was enzymatically capped. According to the protocol, the mRNA was dissolved in an aqueous buffer and combined with GenVoy ILM (Precision Nanosystems, Vancouver, BC, Canada) at a flow ratio of 3:1 through a microfluidic mixer (Precision Nanosystems, Vancouver, BC, Canada). The solvent was removed by centrifugation at 2000× *g* using a 100 kDa ultrafiltration tube (Milipore, Billerica, MA, USA) and the size and PDI of mRNA-LNP was measured by dynamic light scattering (DLS) on a Malvern Zetasizer Nano-ZS (Malvern, Westborough, MA, UK). The concentration of mRNA-LNPs was measured by an Invitrogen’s Quant-iT Ribogreen RNA assay kit (Invitrogen, Eugene, OR, USA).

### 2.3. Denaturing Formaldehyde Gels

The agarose was melted in 24 mL DEPC water and 3 mL 10 × MOPS buffer (Solarbio, Beijing, China). After the agarose solution cooled to 60 ℃, 6 ml of 37% formaldehyde was added to the solution. The gel was placed into a TBE electrophoresis buffer containing 1 × MOPS buffer. The RNA ladder and samples were heated to 70 ℃ for 10 min, and then loaded onto the gel.

### 2.4. mRNA Transfections

HEK293T cells were seeded in 12-well plates at 100,000 cells/well, and then transfected with 2 μg mRNA using the TransIT^®^-mRNA transfection kit (Mirus, Madison, WI, USA). Six hours later, the medium was replaced with DMEM supplement with 3% FBS.

### 2.5. Western Blotting

After 24 h, the cells transfected with mRNA were collected and incubated in lysis buffer (Beyotime Biotechnology, Shanghai, China). The lysate was run on a 10% SDS-PAGE and transferred to a nitrocellulose membrane. The membrane was incubated with the JEV E mAb 1H10 and blots were developed using ECL reagents (Thermo Fisher Scientific, Waltham, MA, USA).

### 2.6. Animal Experiments

The 6-week-old C57BL/6 mice were purchased from the Laboratory Animal Center of Huazhong Agricultural University, Wuhan, China. Mice were randomly assigned into 4 groups: DMEM group (*n* = 16); SA14-14-2 group (*n* = 16); LNP group (*n* = 16); prM-E-mRNA group (*n* = 16). For vaccinations, the mice in prM-E-mRNA and LNP groups were injected intramuscularly with 15 μg LNP-mRNA or LNP-empty, respectively, and boosted with the same dose 21 days later. Mice in SA14-14-2 and DMEM groups were immunized only once with a 100 μL SA14-14-2 attenuate vaccine or the same volume of DMEM, respectively. The serum was collected at day 21 or day 42 and all mice were challenged with 1 × 10^6^ PFU of JEV P3 strain. The experimental animal protocols were approved by The Scientific Ethics Committee of Huazhong Agriculture University (HZAUMO-2021-0003).

### 2.7. Flow Cytometry

Three weeks after the boost vaccination, three mice were randomly selected from each group for spleen isolation. The spleens were ground on a 40-μm-pore-size cell strainer. Red cell lysis solution was added to the cell suspension for 5 min, then PBS was added to stop the lysis. The solution was centrifuged at 1600× *g* rpm for 5 min, the supernatant was removed, and the DMEM supplement with 3% FBS was added to resuspend. In total, 106 cells were incubated with antibodies for 30 min in the dark. Cells were washed twice with PBS, resuspended in 1% paraformaldehyde and detected on the machine.

### 2.8. Plaque Reduction Neutralization Test

BHK-21 cells were seeded in 24-well plates at 50,000 cells/well a day early. Serial dilutions of heat-inactivated serum from mice were incubated with ~100 PFU of JEV for 90 min at 37 °C, then inoculated onto monolayer cells at 37 °C for 60 min. Subsequently, the supernatants were removed and incubated with sodium carboxymethyl cellulose (Sigma, St. Louis, MO, USA) containing medium supplemented with 3% FBS. Five days later, the cells were fixed with 10% formaldehyde stained with a crystal violet solution. Plaque numbers were recorded, and a neutralization titer was calculated according to the Reed–Muench method.

### 2.9. Enzyme-Linked Immunosorbent Assay

Sera were harvested from mice and the level of IFN-γ in the sera was measured using ELISA kits (Abclonal, Wuhan, China), following the instructions.

### 2.10. RNA Extraction and Quantitative Real-Time PCR

After the mouse brain tissue was isolated, PBS was added for grinding, and the supernatant was collected by centrifugation, followed by RNA extraction. Total RNA was reverse transcribed by the ABscript II cDNA First Strand Synthesis kit (Abclonal, Wuhan, China). Then, a qRT-PCR was performed using SYBR Green First qPCR Mix (Abclonal, Wuhan, China) and the QuantStudio 6 Flex PCR system (Thermo Fisher Scientific, Waltham, MA, USA). The primers are listed in Table 1.

### 2.11. Data Analysis and Statistics

All experiments were performed at least three times under similar conditions. Analyses were conducted using GraphPad Prism Software 8.0. *p* < 0.05 was considered significant.

## 3. Results

### 3.1. Construction of the mRNA Vaccine Encoding JEV prM-E Proteins

A vector encoding Japanese encephalitis virus prM-E protein with a signal peptide (MRGGNEGSIMWLASLAAVIACAGA) was constructed, which was designed with reference to a West Nile virus prM-E DNA vaccine [18]. The antigen gene was flanked by two previously used untranslated regions [19], and a poly-A tail (Figure 1A) and the prM-E-mRNA were synthesized using T7 RNA polymerase where the UTP was replaced by 1-Methylpseudo-UTP, and the mRNA was then capped enzymatically. The integrity of the mRNA was assessed by agarose gel electrophoresis (Figure 1B). To examine the expression of the prM-E protein, HEK-293T cells were transfected with the mRNA, and the cell lysate was collected for immune-blot assay at 36 h post transfection. The result showed that JEV prM-E proteins were efficiently expressed in HEK-293T (Figure 1C). The modified mRNA was encapsulated into a lipid nanoparticle (LNP), the particle size of which was approximately 80 nm (Figure 1D).

### 3.2. Immunogenicity of the JEV prM-E-mRNA Vaccine

To assess the immunogenicity of the JEV prM-E-mRNA vaccine, 6-week-old C57BL/6J mice were immunized twice with 15 μg LNP-mRNA or LNP-empty, and the SA14-14-2 live-attenuated vaccine was used as the control (Figure 2A). Serum was collected at day 21 or 42 post immunization and the levels of neutralizing antibodies were determined by a plaque reduction neutralization test (PRNT). As expected, mice vaccinated with SA14-14-2 and JEV prM-E-mRNA showed increased levels of neutralizing antibodies, with a PRNT50 titer reaching ~1:200 and 1:100, respectively (Figure 2B). In contrast, no neutralizing antibody was detected in the sera of mice in the DMEM and LNP groups. To evaluate the cellular immunity elicited by the mRNA vaccine, three mice from each group were randomly selected for spleen lymphocyte isolation, and the changes of CD3^+^CD4^+^ and CD3^+^CD8^+^ T cells were measured by flow cytometry. The results showed that the responses of the CD8^+^ T cells were strongly elicited after mRNA vaccination, while the CD4^+^ T cells were not obviously changed (Figure 2C). We also detected the level of IFN-γ in serum by an ELISA kit with data showing that the mRNA vaccine could increase the IFN-γ concentrations in serum compared with the LNP group. Likewise, the SA14-14-2 group showed a similar trend of IFN-γ production (Figure 2D). The above results indicated that prM-E-mRNA vaccines successfully induced functional neutralizing antibodies against JEV and stimulated strong cellular immunity responses.

### 3.3. Protective Effect of the mRNA Vaccine against JEV 

To assess the protective efficiency of the mRNA vaccine against JEV, the above four groups of mice were challenged with 1 × 10^6^ PFU JEV P3 strains at 42 days after the first immunization. The results showed that vaccination with the mRNA vaccine and SA14-14-2 live-attenuated vaccine protected mice from weight loss and improved behavioral signs. However, a significant reduction in body weight and obvious clinical symptoms were observed in the DMEM and LNP group starting from day 5 post infection (Figure 3A,B). In total, 100% of mice immunized with prM-E-mRNA or SA14-14-2 survived, while the mice in LNP or DMEM control group all died within 11 days after the JEV challenge (Figure 3C). To further confirm the protective effect of the mRNA vaccine, the viral loads in the mice brains were measured. The results of a plaque assay and RT-qPCR indicated that no infectious virus and viral RNA were detected in the brain tissue of the mRNA vaccine- and attenuated vaccine-immunized mice on day 5 post-infection (Figure 3D). These results suggest an effectively protective activity of the mRNA vaccine against JEV.

### 3.4. The mRNA Vaccine Immunization Attenuates JEV-Caused Neuroinflammation Response in Mice

Given that the mRNA vaccine protected mice from a lethal challenge with JEV, we next evaluated whether the vaccine could prevent the associated inflammatory response in the mouse brain. On day 5 after the JEV challenge, the brain tissues of three mice from each group were collected. The expression of inflammatory cytokines, including IL-6, CCL2, CCL5 and TNF-α, were measured by RT-qPCR. The results showed that JEV infection caused a massive expression of inflammatory cytokines, and vaccination could significantly reduce the production of inflammatory cytokines (Figure 4A). In addition, the JEV caused the pathologic changes and gliosis were examined by HE and IHC staining respectively. As shown in the HE staining results, in the brains of mice inoculated with LNP or DMEM, blood vessels were highly congested, and a large number of infiltrated inflammatory cells and vascular sleeves were observed (Figure 4B). In addition, nodules of glial cells, neuronal vacuolar necrosis and nuclear metachromasia were also shown in these two groups. However, no obvious pathological changes were observed in the brains of mice in the mRNA vaccine and attenuated vaccine groups. The IHC results revealed that microglia and astrocytes in unvaccinated mice showed an activated state of antennae-like expansion and a darker brown cytoplasm (Figure 4C). In contrast, mice vaccinated with mRNA or SA14-14-2 vaccines showed no obvious changes in the morphology of glial cells. The integrated option density analysis of the images also confirmed the observation. Taken together, these results suggest that the mRNA vaccine could largely attenuate a JEV-induced inflammatory response in mice brains.

## 4. Discussion

Japanese encephalitis (JE) is a mosquito-borne viral disease which is endemic in many countries in Asia and the Western Pacific and four percent of symptomatic cases progress to serious disease [20]. It was reported that 50,000–175,000 children aged 0–14 years suffered from JE each year [21]. JEV is generally considered to be a critical factor in reproductive failure in swine. As an amplifying host for JEV, swine is the core part during the epidemic [22]. Several studies have demonstrated that the swine–mosquito–swine transmission is promoted when high viremia is developed in JEV-infected domestic swine [23,24]. Therefore, vaccination is equally significant for swine, which could not only protect swine from possible reproductive disorders but also break the transmission cycle, reducing the negative impact on human health [25]. Over the past years, SA14-14-2 vaccines for humans have been broadly used in many countries where JE is endemic, as well as inactivated or live-attenuated vaccines for swine [26]. New inactivated vaccines, recombinant protein subunit vaccines and DNA vaccines against JEV were reported in recent years [7,8,27]. However, there are no related reports on mRNA vaccines of JE. Compared with the above several vaccines, mRNA vaccines possess unique advantages: the simple processing condition, the great safety and the potential to improve immunogenicity with the modification technology. Since the COVID-19 pandemic, the development of an mRNA vaccine platform has been further enhanced, and Moderna and Pfizer’s mRNA vaccines showed effectiveness and safety in human trails, which indicated that mRNA vaccines may be expected to be a new method to replace traditional vaccines [28,29].

E and prM proteins of JEV are two structural proteins, from which prM is cleaved into the M protein during the process of virions’ maturation by furin, that recognize the cleavage site. The prM protein is an important synergistic component of virus-induced protective immunity and promotes the correct folding of the envelope protein [30,31]. As a glycoprotein, the E protein is the main antigenic protein which induces neutralizing antibodies and mediates membrane fusion [32]. DNA vaccines coding prM and E proteins appear to provide more effective protection than constructs expressing the E protein alone [33,34].

In this study, mRNA encoding the prM and E proteins of JEV was successfully expressed in the cells. The mRNA vaccine against JEV manufactured by modifying mRNA and using the LNP technology platform induced effective neutralizing antibodies and activated cellular immunity in mice. As in the case with the SA14-14-2 group, the mice vaccinated with the mRNA-prM-E vaccine all survived in the JEV challenge. Nevertheless, the neutralizing titer of the mRNA vaccine after secondary immunization was lower that of the live-attenuated vaccine, which may be due to the level of protein expression. Therefore, we need to further modify the mRNA to promote the efficiency of translation in follow-up studies. In addition, no significant increase in CD4^+^ T cells was seen in the spleen of the mice that received the mRNA vaccine. This may also explain the phenomenon that the neutralizing antibodies’ level of the mRNA vaccine is not as high as the SA14-14-2 vaccine, since the Th2 cells mainly contribute to humoral immunity. On the other hand, CD8^+^ T cells and the IFN-γ level in serum were notably enhanced, suggesting that the mRNA vaccine could stimulate cell immunity in mice, which may also protect mice against the JEV challenge.

In this study, immunization of the attenuated vaccine enhanced the CD4^+^ T cell responses significantly (Figure 2C), which may explain the high titer of neutralizing antibodies detected in the SA14-14-2 group. This finding is consistent with a previous study which reported that the CD4^+^ T cells stimulated by the SA14-14-2-attenuated vaccine are sufficient to give complete protection to mice, depending on the antibody response [35]. It has been well known that an mRNA vaccine could stimulate both CD4^+^ and CD8^+^ T cell responses [36]. In our study, the mRNA vaccine of JEV showed a higher proportion of CD8^+^ T cells than CD4^+^ T cells, which may be dependent on the antigen properties.

The prM-E protein in the JEV p3 strain belonging to Genotype III (GIII) was used for the mRNA vaccine design in this study. Except for GIII, JEV strains in other genotypes such as GI and GV are also prevalent in Asia. Therefore, the mRNA vaccines that confer a protective effect for different genotypes of JEV are required. Given that the amino acid homology of the E protein in JEV of GIII and GI is higher than that of G III and GV, the mRNA-LNP vaccine designed based on the GIII strain is supposed to be more effective on JEV GI strains than GV strains. This speculation is supported by the report which found low levels of neutralizing and protective antibodies against the Chinese GV isolate XZ0934 strain in SA14-14-2-derived live-attenuated vaccine immunized mice [37]. Certainly, it is important to determine the protective effect of our mRNA vaccine on other genotypes of JEV in future studies.

In summary, we described the generation of an mRNA vaccine coding prM and E proteins against JEV, which could elicit both humoral and cellular immune responses and protect mice from a lethal JEV challenge. This study may provide new clues for the development of more efficient and safer JEV vaccines.

## Figures and Tables

**Figure 1 viruses-14-01121-f001:**
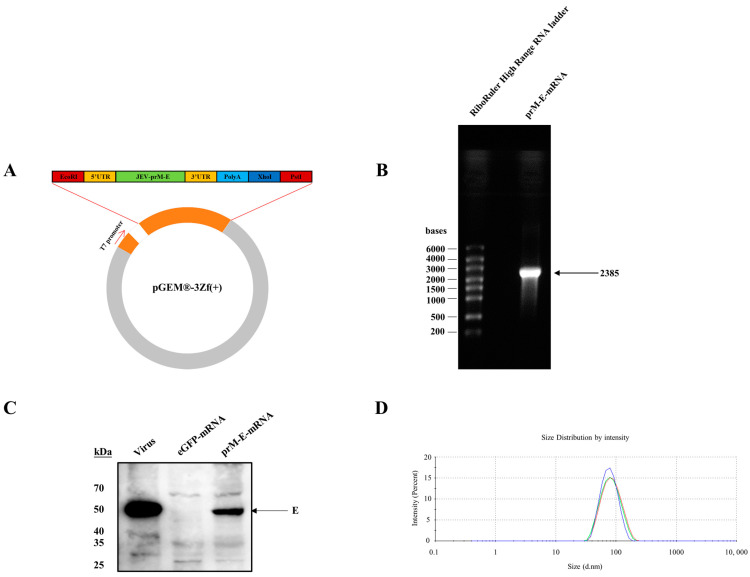
Construction and expression of mRNA-prM-E. (**A**) Characterization of vectors for in vitro transcription. **(B**) mRNA-prM-E was run in the 1% formaldehyde denaturing gel (RNase-free). (**C**) HEK-293T cells were transfected with the mRNA-prM-E, and lysate was analyzed by Western blotting with a monoclonal antibody 1H10 against the JEV E protein. The pcDNA3.1 plasmid encoding prM-E and JEV were used as the positive controls. (**D**) The particle size of the LNP-mRNA was measured three times using a Zetasizer spectrometer indicating an average size of approximately 80 nm.

**Figure 2 viruses-14-01121-f002:**
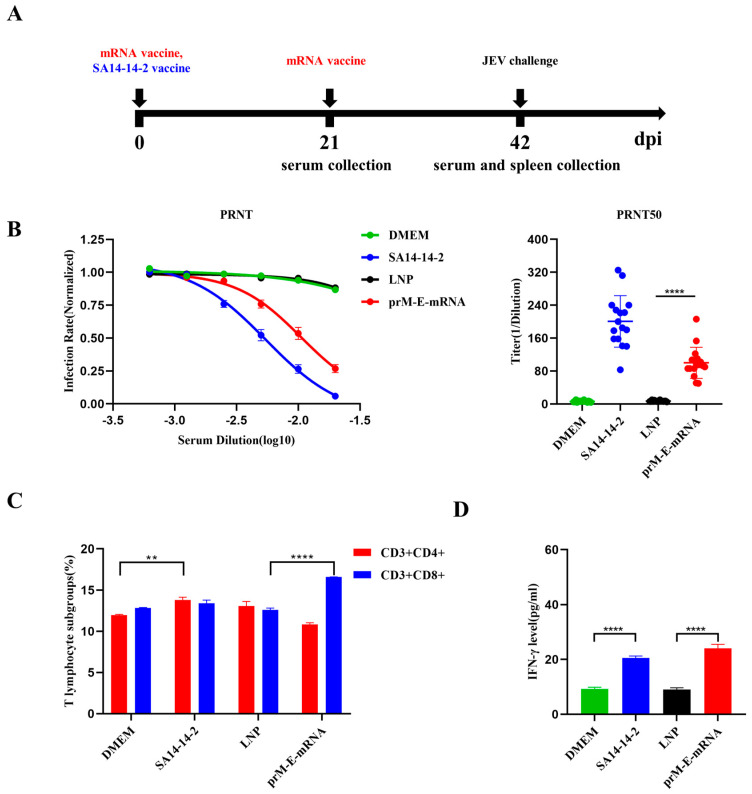
Immune responses in C57BL/6 induced by the mRNA-prM-E vaccine. (**A**) Mice were intramuscularly injected with 15 μg of mRNA-prM-E or SA14-14-2 live-attenuated vaccine. The sera of mice in SA14-14-2 group and DMEM group were collected on day 21 while the sera of mice immunized with prM-E-mRNA or LNP were collected on day 42. (**B**) Serial dilutions of serum were tested for neutralization activity by PRNT assay. Neutralization curves (left panel) and the PRNT50 (right panel) are shown. Each experiment was independently repeated three times and PRNT50 was calculated by the Reed–Muench method (*n* = 16, ** *p* < 0.01; **** *p* < 0.0001). (**C**) The percentages of CD8 ^+^ and CD4 ^+^ T cells in spleen were analyzed by flowcytometry. Data are represented as mean ± SEM of 3 spleens from 3 mice from each group. (** *p* < 0.01; **** *p* < 0.0001). (**D**) The IFN-γ level in sera was measured by ELISA. Data are represented as mean ± SEM. (*n* = 16, ** *p* < 0.01; **** *p* < 0.0001).

**Figure 3 viruses-14-01121-f003:**
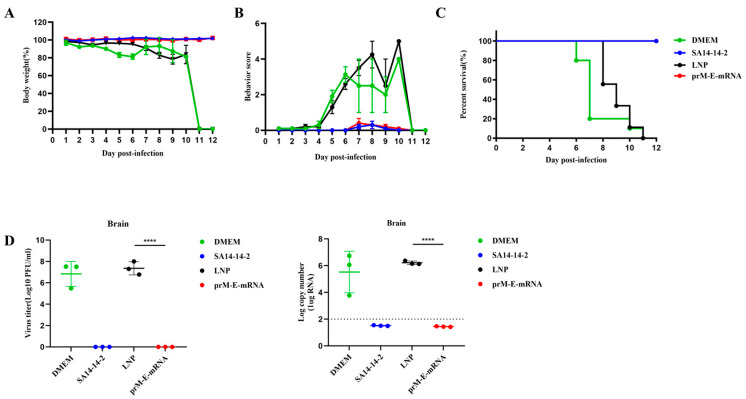
Protective efficacy against JEV in C57BL/6 after immunization. Body weight changes (**A**), behavioral score (**B**) and survival rate (**C**) were recorded after JEV P3 strain challenge. Survival rate is displayed as Kaplan–Meier survival curves and behavioral changes from normal to severe are represented on a scale of 0 to 5 (*n* = 10). (**D**) Viral loads in mice brain were detected by plaque assay and qRT-PCR at day 5 after JEV-infection. Data are represented as mean ± SEM. (*n* = 3, **** *p* < 0.0001).

**Figure 4 viruses-14-01121-f004:**
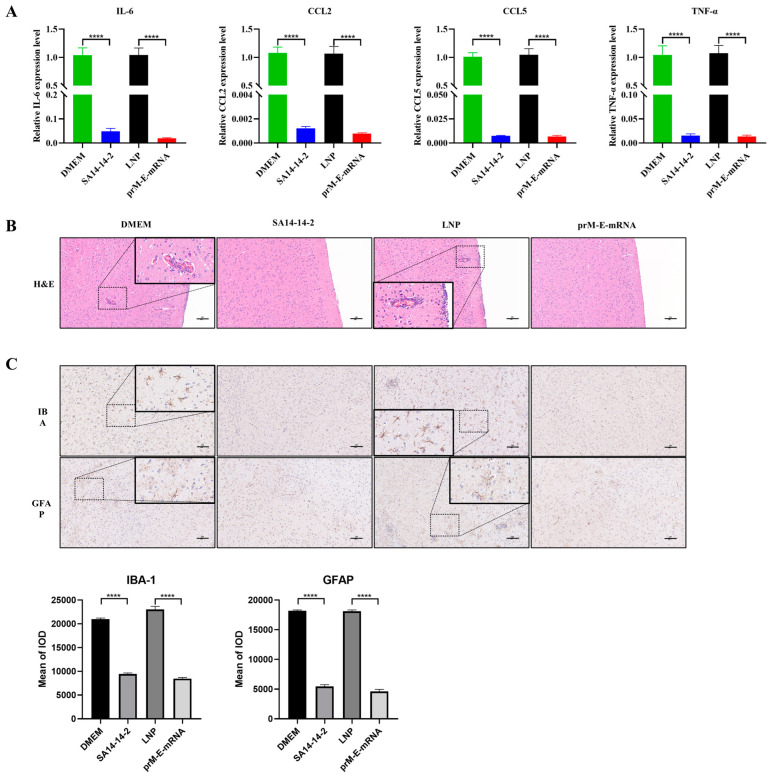
The mRNA vaccine protects against a JEV-induced inflammatory response in mice brain. (**A**) The mRNA expression levels of inflammatory cytokines (IL-6, CCL-2, CCL-5, TNF-α) in brain tissue lysates were quantified by qRT-PCR. (**B**,**C**) The activation of the pathological changes and glial cells in brain tissue of mice were analyzed by H&E staining and IHC. Integrated option density analysis of three visual fields of slides from each group was performed to quantify the results of immunohistochemical staining (the lower panels). Scale bar = 100 μm. Data are represented as mean ± SEM. (*n* = 3, **** *p* < 0.0001).

**Table 1 viruses-14-01121-t001:** Primers used in this study.

Name	Sequence (5′-3′)
prME-F	AGAGCGGCCGCTTTTTCAGCAAGATTAAGCCGCCACCATGA GAGGAGGAAATGAAGGCTCAA
prME-R	GGGGTACCTCAAGCATGCACATTGGTCGCTAAGAAC
mTNF-α-F	TGTCTCAGCCTCTTCTCATTCC
mTNF-α-R	TTAGCCCACTTCTTTCCCTCAC
mIL-6-F	AATGAGGAGACTTGCCTGGT
mIL-6-R	GCAGGAACTGGATCAGGACT
mCCL2-F	CGGCGAGATCAGAACCTACAAC
mCCL2-R	GGCACTGTCACACTGGTCACTC
mCCL5-F	TGCCCACGTCAAGGAGTATTTC
mCCL5-R	AACCCACTTCTTCTCTGGGTTG
JEV-F	TGGTTTCATGACCTCGCTCTC
JEV-R	CCATGAGGAGTTCTCTGTTTCT

## Data Availability

All data generated and analyzed in this research are included in the article.

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
