# Peer review of "Protective Immune Responses Induced by an mRNA-LNP Vaccine Encoding prM-E Proteins against Japanese Encephalitis Virus Infection"

_viruses, 2022, doi:10.3390/v14061121_

Round 1

Reviewer 1 Report

The authors have developed a modified mRNA vaccine expressing JEV proteins, which has been proved to be functional in inducing cellular immunity against virus challenge and comparable/better than the live attenuated vaccine. The results can support their conclusions, but the manuscript can be further improved after addressing the comments below.

Comments for authors

  1. In figure 2B & 2D, the authors should specify on which day the sera were tested, Day 21 or 42?
  2. It would enhance the quality of the paper by testing if the increase in splenic CD8 T cells in Fig. 2C is antigen specific. For example, antigen re-stimulation and intracellular staining in vitro or tetramer staining would achieve that.
  3. In Fig.4, the authors should specify on which day these brain samples were tested.
  4. It would be better to show the comparison of pathological scores in Fig.4B&C, instead of representation images only.
  5. In line 183, the authors claim, “the results indicated prM-E-mRNA vaccine successfully induced functional neutralizing antibodies against JEV”. However, there is no direct evidence in the manuscript to support it, so the authors can either edit the sentence or include other results.

Reviewer 2 Report

An mRNA-lipid nanoparticle (LNP) vaccine encoding prM-E proteins of Japanese encephalitis virus (JEV) was developed and assessed its protective efficacy in 6-week-old mice. The mRNA-LNP showed an increase in antibody titer after being given a booster dose. The vaccine showed equivalent protective efficacy as observed in the SA-14-14-2 attenuated vaccine. The mRNA-LNP tends to augment the CD8 population in the spleen and interferon-gamma levels in serum. Overall, the experiments are well planned and executed with proper control.

Minor comments:

1. In Fig 2C, the difference in CD4 level between DMEM and SA14-14-2 group is marginal. Data from more mice may need to conclude.

2. It is not clear how the author decides the vaccine dose.

3. What is the antibody titer generated after being given 1st dose of the vaccine?

4. Will the mRNA-LNP vaccine be effective against all JEV genotypes?

5. Why is there an increase in the CD8 population? Generally, subunit vaccines fail to develop CTL response.

6. The data will be more promising if they provide evidence of the presence of memory B cells and their stability?

Reviewer 3 Report

Manuscript Number: viruses-1698257

The manuscript by Chen et al. entitled “Protective immune response induced by an mRNA-LNP vaccine encoding prM-E proteins against Japanese encephalitis virus infection” is a relevant study. It was reported that a modified mRNA vaccine expressing the prM and E proteins of the JEV-P3 strain was designed and formulated with LNP-tested immune responses. It was used to assess its efficacy, where SA14-14-2 live attenuated vaccine was used as the control. Japanese encephalitis (JE) virus is the leading cause of vaccine-preventable encephalitis in Asia and the western Pacific. Vaccination is a safe and effective way to reduce your chance of catching JEV. The research paper provides a great insight into the role of induced neutralizing antibodies against JEV and protecting mice from JEV-challenge and provides a firm base for further development of mRNA-based vaccine direction.

This approach has potential implications for the development of a JEV mRNA-based vaccine and may also have applications for other viruses and their pathogeneses. This approach also demonstrates interesting biological phenomena and prophylactic JEV vaccine responses in mouse models. The paper is well written, and the experimental design is good, including all the necessary control groups. However, the scope of this paper is too broad, and the data may be sufficient to conclude.

The paper would be acceptable after addressing some of the issues described below.

  1. Functional studies may require enumerating the epitope binding predictions of LNP-mRNA as a candidate vaccine.
  2. It would have been nice to see an explanation for the differences in terms of CD8+ and CD4+ T cell immune responses for mRNA vs. live attenuated vaccine. T cell-dependent/independent subsets (CD4 vs. CD8) would improve the immune response LNP-based mRNA vaccine against JEV.
  3. For the neutralization assay should contain diluted irrelevant immune sera for assay control. Most of the flavivirus can be neutralized by non-immune mouse serum, which should be considered in neutralization assays.
  4. Also, the authors do not adequately include the correct references of other vaccine platforms.

Overall summary

  • Are the objectives of the study clearly articulated, with a clear testable hypothesis stated? Yes
  • Is the study design appropriate to address the stated objectives? Yes
  • Is the population clearly described and appropriate for the hypothesis being tested? Yes
  • Is the sample size sufficient to ensure adequate power to address the hypothesis being tested? Yes
  • Were correct statistical analyses used to support conclusions? Yes
  • Are there concerns about ethical or regulatory requirements being met? Yes.

Round 2

Reviewer 1 Report

The authors have addressed my comments in their initial submission, although some assays cannot be done due to the COVID shutdown in Shanghai, unfortunately. Even so, the authors have done sufficient work to demonstrate their conclusion. Therefore, I would like to accept this manuscript.